# Microplastic-Free Microcapsules to Encapsulate Health-Promoting Limonene Oil

**DOI:** 10.3390/molecules27217215

**Published:** 2022-10-25

**Authors:** Daniele Baiocco, Zhibing Zhang

**Affiliations:** School of Chemical Engineering, University of Birmingham, Birmingham B15 2TT, UK

**Keywords:** microencapsulation, microplastic-free, plant-based, coacervation, micromanipulation

## Abstract

Fast-moving consumer goods (FMCG) industry has long included many appealing essential oils in products to meet consumers’ needs. Among all, the demand for limonene (LM) has recently surged due to its broad-spectrum health benefits, with applications in cosmetic, detergent, and food products. However, LM is extremely volatile, hence has often been encapsulated for a longer shelf-life. To date, mostly non-biodegradable synthetic polymers have been exploited to fabricate the microcapsule shells, and the resulting microcapsules contribute to the accumulation of microplastic in the environment. So far, information on LM-entrapping microcapsules with a natural microplastic-free shell and their mechanism of formation is limited, and there is lack of an in-depth characterisation of their mechanical and adhesive properties, which are crucial for understanding their potential performance at end-use applications. The present research aims towards developing safe microcapsules with a core of LM fabricated via complex coacervation (CC) using gum Arabic (GA) and fungally sourced chitosan (fCh) as shell precursors. The encapsulation efficiency (EE) for LM was quantified by gas chromatography (GC) separation method. The morphology of microcapsules was investigated via bright-field optical microscopy and scanning electron microscopy, and their mechanical properties were characterised using a micromanipulation technique. Moreover, the adhesive properties of the resulting microcapsules were studied via a bespoke microfluidic device fitted with a polyethylene-terephthalate (PET) substrate and operating at increasingly hydrodynamic shear stress (HSS). Spherical core-shell microcapsules (EE ~45%) with a mean size of 38 ± 2 μm and a relatively smooth surface were obtained. Their mean rupture force and nominal rupture stress were 0.9 ± 0.1 mN and 2.1 ± 0.2 MPa, respectively, which are comparable to those of other microcapsules with synthetic shells, e.g., urea- and melamine-formaldehyde. It was also found that the fCh-GA complexed shell provided promising adhesive properties onto PET films, leading to a microcapsule retention of ~85% and ~60% at low (≤50 mPa) and high shear stress (0.9 Pa), respectively. Interestingly, these values are similar to the adhesion data available in literature for microplastic-based microcapsules, such as melamine-formaldehyde (50–90%). Overall, these findings suggest that microplastics-free microcapsules with a core of oil have been successfully fabricated, and can offer a potential for more sustainable, consumer- and environmentally friendly applications in FMCGs.

## 1. Introduction

Limonene (LM) is one of the most available hydrophobic monocyclic monoterpenes in the vegetable kingdom [1]. As with other phytochemicals, it naturally occurs in many blends of essential oils (EOs), which provides them with an agreeable piny-lemon turpentine-like scent [2]. Over the years, its proven harmlessness for humans and potential health promoting properties have brought LM towards increasingly high acceptance, especially for applications in cosmetics and pharma-nutraceutical formulations, as well as agri-food & beverage [3]. Specifically, LM’s pleasant odour has long been playing a crucial role in perfume and personal care products [4]. Similarly, LM has been extensively exploited as food & beverage aid by manufacturers to boost consumers’ satisfaction towards building their lifelong relationship [5]. More importantly, LM provides not only a delightful sensation, but also has demonstrated broad-spectrum therapeutic benefits encompassing anti-inflammatory, antioxidant, analgesic, anti-hyperalgesic, immunomodulatory, antinociceptive, anticancer, antigenotoxicity, antidiabetic, antiviral, antiallergic, antimicrobial and gastroprotective effects [1,6,7]. Moreover, LM has demonstrated herbicidal and anthelmintic activities [8], hence may offer a novel, safer and greener bio-alternative to highly popular synthetic pesticides [9], which are typically delivered using non-degradable microplastic-based carriers [10]. However, some limitations are associated with LM due to its extreme volatility and thermo-chemical instability [11]. LM is prone to evaporating and may easily undergo degradation/autoxidation when exposed to hostile environment, such as heat, oxygen, and light [1]. Therefore, LM has often been encapsulated to circumvent such shortcomings, hence achieving a longer shelf-life of products containing LM itself. To date, researchers have mainly focused on using non-biodegradable synthetic polymers to fabricate the microcapsule shells, such as highly performing melamine-formaldehyde, [12,13], but the resulting microcapsules contribute to the accumulation of persistent microplastic in the environment [14]. 

Recently, animal-based ingredients, such as seafood-derived chitosan [15] and bovine/swine gelatine [16], have proven effective to form the microcapsule shells around emulsified LM droplets. More specifically, researchers have investigated the potential of animal sourced chitosan (a-Ch) as an encapsulating ingredient [17,18]. Interestingly, a-Ch is a cationic polyelectrolyte derived from shellfish exoskeleton which has proven effective in encapsulating lipophilic ingredients, such as LM, leading to multifunctional microcapsules with a potential for personal care products [19]. Notwithstanding, recent research has evidenced the ubiquitous presence of microplastics in blood as well as animal tissues, and seashells indeed [20,21]. This type of contamination may also pose a severe risk to humans once microplastic-integrated seafood has entered the human food chain [22]. Similarly, microcapsules with a-Ch based shell may occur with microplastics, which are then non-fully biodegradable. Accordingly, there have been increasing regulations on restricting the use of synthetic solid polymers and/or environment polluting microplastic residues in the microcapsule shells [23]. Alongside, there is a growing demand from end-use consumers for genuinely plant-based ingredients to tackle the risk associated with animal sources (e.g., protopathic diseases), and to meet their personal and religious beliefs [5]. As a step forward to comply with the foregoing requirements, noteworthy efforts are being undertaken by researchers to shift towards using more sustainable microplastic-free materials as the shell precursors. In our previous works, we reported the potential of fungally fermented chitosan (fCh) which had proven effective to form shells around model perfume (i.e., hexyl salicylate) and food flavour (i.e., L-Carvone) oils by complex coacervation [5,10,24]. When compared to crustacean sourced chitosan, fCh occurs with a relatively lower chitin content, thereby resulting in a larger number of covalently decorated *β*-glucan branches [25]. Therefore, fCh may benefit from enhanced conformational extensibility, as well as a more pliable structural scaffolding [24]. Moreover, unlike a-Ch, fCh requires no aggressive acid-driven treatment to eliminate calcium carbonate and other mineral impurities.

To date, information on LM-entrapping microcapsules with a natural microplastic-free shell and their mechanism of formation is very limited, and there is lack of an in-depth characterisation of their mechanical and adhesive properties, which are crucial for understanding their performance at potential end-use applications, such as in textiles [26] and laundry detergents [27]. Although Souza et al. [15] have reported using Ch microcapsules to immobilise onto cellulose fabrics, their study only focused on the kinetic release of the active ingredient (LM). Similarly, Sánchez-Navarro, Pérez-Limiñana, Arán-Ais and Orgilés-Barceló [26] have described the potential of melamine-formaldehyde and gelatine-carboxymethylcellulose microcapsules retaining LM to incorporate via simple hydrodipping-impregnation procedure into footwear linings and insoles, respectively, which may have a beneficial effect against sweating. Nonetheless, to the best of our knowledge, no extensive study on the mechanical and adhesive properties of vegetable chitosan based LM-entrapping microcapsules has been conducted so far. Therefore, the present research aims towards (*i*) developing safe microcapsules with a core of LM, fabricated via complex coacervation using gum Arabic (GA) and fCh as the shell precursors, (*ii*) characterising their mechanical properties using an advanced micromanipulation technique while being cognisant of the potential effect of specific fragrance oils (LM) on the microcapsule mechanical strength in comparison with the state of the art, and (*iii*) studying their adhesive properties using a bespoke microfluidic device fitted with a smooth polyethylene terephthalate (PET) substrate and operating at increasingly hydrodynamic shear stress and different environmental pHs to mimic real fabric washing and/or human sweat conditions. 

## 2. Results & Discussion

### 2.1. Morphology

Figure 1 displays the digitalised photomicrographs of LM-entrapping microcapsules fabricated by complex coacervation. As can be seen, suspended microcapsules with a yolk-white structure were obtained. Specifically, the microcapsules exhibited individual oil droplets encircled by fCh-GA polymer shells with a slightly elongated or eye-like shape. As previously observed by Leclercq, Harlander and Reineccius [16] from gelatine based microcapsules with a core of LM, this eye-shaped configuration may be ascribable to fast stirring while inducing coacervation, thereby triggering the alignment of the droplets/microcapsules with the flow pattern within the agitated vessel. Moreover, Baiocco, Preece and Zhang [10] have elucidated that some excess shell-forming material could be deformed around the oil droplets by the agitation during the development of the shell, hence generating the eye shape, as reported for microcapsules with a core of hexyl salicylate within a fCh-GA based shell. Interestingly, the suspended microcapsules did not tend to agglomerate. The outline of each individual microcapsule with a relatively smooth surface was clearly visible. Moreover, no free oil droplets were found, which might provide some qualitative information on the encapsulation efficiency of microcapsules. However, this does not mean that the whole volume of oil (LM) used (45 mL) was entirely encapsulated within the coacervate shells. LM tends to evaporate quickly at ambient conditions (p_sv,LM_ (20 °C) ~0.2 kPa), and even quicker during homogenisation due to the dissipated heat leading to an increased temperature (~35 °C). In other words, some emulsified oil droplets originally occurring in bulk might have been evaporated prior to being caught up by a coacervate network. Notwithstanding, a proper quantification of the encapsulation efficiency and payload of microcapsules by UV-Vis is required.

SEM micrographs of LM-microcapsules are displayed in Figure 2. Individual and clusters of microcapsules with a spherical shape and relatively smooth surface were observed (Figure 2A). The surface topography of microcapsules suggests the presence of surface wrinkles which may be a consequence of the high vacuum during SEM. Moreover, several dark spots were also discernible. Since they appeared blurred, they might be located at a significant depth from the shell surface. Most likely, these dark spots might indicate the presence of subshell reservoirs of LM (Figure 2B). Partly incomplete microcapsules and polymeric debris were also found (Figure 2C). Hemispherical microcapsules with a core-shell structure were evident. SEM micrographs also enabled to estimate the shell thickness of LM-microcapsules preliminarily, which was around 0.5–1.5 µm. However, this estimation was not accurate, thus TEM was performed.

### 2.2. Particle Size Distribution

Figure 3 displays the size distribution of LM microcapsules in the range of 9–112 μm (SPAN = 1.21), with a mean Sauter (surfaceweighted *D*_[3,2]_) diameter of 38 ± 2 μm. As shown, a single-peak distribution suggested that clusters of microcapsules were unlikely in suspension. Moreover, statistical analysis showed that a log-normal distribution can be fitted to the size distribution data with 95% confidence. These results also confirmed that the size of microcapsule was suitable for their inoculation into the flow chamber (inner height of the flow chamber 280 µm) as described in Section 3.3.8, without any risk for clogging of the passageways of the microfluidic device.

### 2.3. Encapsulation Efficiency and Payload

LM-microcapsules were assayed for EE and payload via GC. The EE and payload were 44 ± 3% and 29 ± 2%, respectively. These results appeared in good agreement with the values which had been obtained from microcapsules with a core of hexyl salicylate (HS), regardless of the different analytical technique used (i.e., UV-Vis for HS-microcapsules) [10]. Interestingly, the standard errors associated with both EE and payload from LM-microcapsules were approximately three-fold lower than those of HS microcapsules (EE_%,HS_ = 47 ± 11% and P_%,HS_ = 40 ± 7%). This finding is probably ascribable to the different level of accuracy and sensitivity between these two analytical techniques. As reported in literature, GC is often preferred for the identification, quantification, and separation of specific terpenic analytes, such as limonene [28]. Although highly accurate, GC is time consuming and rather expensive, whereas UV-Vis is more rapid and versatile. 

Our previous studies have demonstrated that the static leakage (in air) of terpenic compounds from fCh-GA coacervate shells was negligible after 50 days [5]. Similarly, another model oil (i.e., hexyl salicylate) encapsulated within a fCh-GA shell showed no significant oil leakage in water after one month [10]. Having said that, limonene laden microcapsules may be expected to perform similarly, which can be appealing for many fast-moving consumer goods (FMCGs) requiring a prolonged retention of the active. The LM leakage rate under different environmental conditions remains to be measured directly, which is future work.

### 2.4. Mechanical Properties

Figure 4A displays the rupture force of LM-microcapsules as a function of diameter. Interestingly, the rupture force increased with diameter, which is consistent with the trend of microcapsules with a similar nature (HS microcapsules [10]) and/or different chemistry (i.e., melamine-formaldehyde [27]). Similarly, the nominal rupture stress (Figure 4B) decreased with diameter, whereas the displacement at rupture increased with diameter (Figure 4C). Although the trends were found to be in good agreement with those from the other microcapsules, their average values differed significantly. The average rupture force of LM-microcapsules (0.93 ± 0.10 mN) was approximately half of that of HS-microcapsules (2.0 ± 0.1 mN). Interestingly, this result was approximately one third of the maximum value of commercially relevant microcapsules (2.66 ± 0.31 mN) with similar size (15–25 µm) and a melamine-formaldehyde based shell with a 10:1 (*w*/*w*) admixture of HB40-kerosene as the core (HB40-microcapsules) [27]. When dealing with the number-based diameter of LM-microcapsules obtained from micromanipulation, its value (*d*_LM_ = 24.4 ± 1.2 µm) appeared to be similar to that of HS-microcapsules (*d*_HS_ = 27.5 ± 1.5 µm) [10]. Thus, the mechanical properties of the two types of microcapsules may be compared directly. Based on the results, LM-microcapsules seem to be naturally weaker and easier to rupture than HS microcapsules. However, the rupture force as such is not sufficient to fully compare the mechanical strength between two types of microcapsules. Their deformation at rupture and nominal rupture stress were thus examined. For two kinds of microcapsules, if one ruptures at a significantly smaller displacement, this is related to a higher structural brittleness of the microcapsule; in contrast, if the other ruptures at a larger displacement, this is associated with a greater structural flexibility and stretchability of the microcapsule. The average displacement at rupture of LM-microcapsules was 3.4 ± 0.4 µm, which is approximately half of that of HS-microcapsules (6.3 ± 3.1 µm). This seems to confirm that LM-microcapsules were ruptured at a smaller deformation on average than HS-microcapsules. Therefore, LM-microcapsules might be structurally more brittle. The average nominal rupture stress was 2.1 ± 0.2 MPa for LM-microcapsules, whereas that of HS-microcapsules were higher (3.6 ± 0.3 MPa). For completion, the average rupture tension, which is normally independent of size, of both types of microcapsules should be verified, which is defined as the rupture force to mean diameter ratio of microcapsules. Accordingly, the rupture tension of LM-microcapsules was determined to be 37.7 ± 3.0 N/m, which is approximately half of that of HS microcapsules (71.6 ± 3.9 N/m), as expected. The relevant mechanical parameters of LM-microcapsules in comparison with those from other types of microcapsules are listed in Table 1.

Overall, these values highlighted that LM-microcapsules were structurally weaker than HS-entrapping microcapsules. Generally, if two microcapsules have similar size, displacement at rupture, and rupture force and hence rupture tension, they then possess comparable mechanical strength. Although the same chemistry and process (i.e., complex coacervation) were applied to fabricate the shell of both LM- and HS-microcapsules, significantly different mechanical properties were found. Therefore, additional considerations should be given. Although LM is deemed to be a relatively stable terpene in microencapsulation [29], the effect of the Span-stabilised emulsion droplets, their chemistry, and the resulting surface interfacial tension on fCh-GA coacervates is still unknown. 

When two immiscible liquid phases (e.g., core oil phase (LM) and polymer-abundant aqueous phase (fCh/GA)) are mixed in an agitated vessel, the combined effects of turbulence (i.e., flow pattern, agitation conditions, impeller and vessel geometry) and their interfacial energy is conducive to the formation of emulsified droplets [30]. However, the resulting emulsions are thermodynamically unstable which may revert back as separate phases promptly [31]. As for our work, surface active agents (SAAs; surfactants) have been broadly utilised to stabilise the disperse phase droplets. SAAs are typically made of amphipathic segments (i.e., hydrophobic and hydrophilic groups) which can adsorb onto the biphasic interface sterically and/or ionically based on their nature [24]. Accordingly, the effective adsorption of SAAs at the droplet interface leads to reducing its interfacial tension and the resulting energy level alignment, thereby stabilising the emulsions [32]. It is widely accepted that the minimisation of surface energy is a necessary but not sufficient criterium towards microencapsulation. Under the appropriate processing conditions (pH, temperature, relative humidity, mechanical agitation, core to shell weight ratio, etc), SAA-assisted energetically lowest lying emulsion systems may offer the most energetically favourable conformations for a coating to form around a droplet [33]. 

To this end, Tasker et al. have reported a possible interrelationship between the interfacial energy of two-phase emulsions stabilised with a cationic class of SAAs (Cetrimonium bromide (C_x_-TAB)) and the morphology of the resulting microcapsules [34]. It is well known that the microcapsule morphology is administered by the interfacial energy balance (IEB) between the phases in the microcapsule and depends upon the wetting conditions between the occurring phases in the system [34,35]. Specifically, the authors employed poly(methyl methacrylate) (PMMA) dispersed in an organochlorine solvent (methylene chloride (MTC)) as the encapsulant polymer and key commercial fragrance oils (i.e., HS and cyclamen aldehyde) as the core ingredient. Based on the mathematical evaluation of the spreading coefficients from the interfacial tensions between the phases present in the final capsule slurry, the microcapsule morphology was tentatively predicted. For encapsulating both fragrance oils, a core-shell structure was predicted and indeed achieved experimentally. This suggested that the polymer-rich phase was sufficient to wet the core phase selectively in MTC. In other words, an energetically favourable configuration between the shell and the core materials along with the appropriate wetting conditions were met, which allowed core-shell microcapsules to form via evaporation of MTC. The evaporation of MTC progressively reduced the polymer solubility within the core phase, thereby triggering the precipitation of PMMA matter onto the core oil droplets [34]. Interestingly, there appeared that microcapsules with similar structures (core-shell morphology) and surface roughness were attained although different encapsulant oils were used. However, the work provided no information on whether such microcapsules (i.e., same shell chemistry and different oil cores) may perform similarly or rather differently with regard to their mechanical properties. 

In our work, sorbitan triolate (Span85^®^) was used as the SAA to stabilise LM microdroplets prior to inducing encapsulation by complex coacervation, as also reported in the literature [10,24,36,37]. As previously discussed, our findings have elucidated that LM-microcapsules were found to be mechanically weaker than the HS-microcapsules, although the same shell chemistry and SAA were employed, as well as the processing conditions. In other words, it seemed that physicochemical interplays between the coacervate shells and oils may not be negligible, which were more evident in the case of LM as the core oil. 

Sharkawy et al. utilised animal sourced chitosan and gum Arabic to fabricate microcapsules with a core of limonene oil by coacervation [36]. Span85 was used as a SAA which led to mostly mononuclear microcapsules, which is in very good agreement with our findings (Figure 1). In addition, the authors encapsulated vanillin essential oil following the same methodology. Both oils led to core-shell microcapsules with similar surface characteristics. In contrast, the use of polyglycerol polyricinoleate (PGPR) as the SAA, under the same processing conditions, resulted in microcapsules with a polynuclear morphology in both cases. Based on their results, the type of the emulsifier (Span85/PGPR) seemed to have a significant influence on the microcapsule morphology, and also on their size, encapsulation efficiency, and barrier properties (i.e., release profile of the core oil through the shell into n-hexane as a receptor medium). However, no insight into the mechanical properties of microcapsules was given. 

Interestingly, the studies presented by Tasker et al. and Sharkawy et al. seemed to suggest that the morphological, surface, and barrier properties might be directly ruled by the IEB between the phases occurring in the microcapsules. Regardless of the encapsulation methodology, core-shell systems are controlled by the relative surface energy of each phase involved towards a total interfacial energy minimisation status [38,39]. In contrast, if the internal core-shell surface energy is relatively large, then a partial/complete detachment between the phases may be favoured [39]. This may tentatively help explain why LM-microcapsules were found to be mechanically weaker than the HS-microcapsules.

This phenomenon may be associated with the interfacial interaction and energy level alignment of the core oil (LM) towards the innermost lining of the fChGA coacervate network. The interfacial energy can vary significantly depending on the specific interaction between a certain solid (e.g., fChGA coacervate network) and a given medium (inner oil phase). The inner core-shell (LM-fChGA) surface energy may be large enough to possibly impair the intermolecular bonds forming the shell network, thereby affecting the resulting thickness and stability of the shell, in turn hindering the overall robustness of the microcapsule. Thus, it cannot be excluded that the LM-fChGA energy level may be significantly higher than that of HS-fChGA due to the nature of terpenic molecules (LM), hence leading to a partial weakening of the solid phase (fChGA) at its intermolecular level. 

Although feasible, the encapsulation of LM seemed less favourable with regard to the microcapsule mechanical properties when compared to that of HS which is an ester oil. Likely, LM is less stable physico-chemically which may have led to the mechanically depleted microcapsule shells.

Despite the key role of the interfacial energy towards determining core-shell interaction and stability [40], as well as the thickness and robustness of the shell, surprisingly, no experimental attempt linking the core-shell IEB to the mechanical properties of microcapsules has arisen so far. It is therefore imperative to conduct further studies to determine the dependency of the microcapsule mechanical properties on the core-shell IEB. This may help towards a new understanding in this field in order to meet the current manufacturing challenges as well as developing novel and highly functional products with enhanced quality at end-use applications. 

**Table 1 molecules-27-07215-t001:** Key mechanical property parameters of LM microcapsules in comparison with other microcapsules reported in literature (Mean ± St. Error).

Mechanical Property Parameter	LM-Microcapsules	HS-Microcapsules [10]	HB40-Microcapsules [27,41,42]
Mean Diameter (μm)	24.4 ± 1.2	27.5 ± 1.5	4.0–24.0 ± 1.0
Rupture Force (mN)	0.93 ± 0.1	2.0 ± 0.1	0.7 ± 0.1– 2.8 ± 0.1
Rupture Tension (N/m)	37.7 ± 3.0	71.6 ± 3.9	-
Nominal Rupture Stress (MPa)	2.1 ± 0.2	3.6 ± 0.3	4.2 ± 0.4
Displacement at Rupture (μm)	3.4 ± 0.4	6.3 ± 3.1	3.5 ± 0.2
Rupture Deformation (%)	14.5 ± 1.0	22.7 ± 1.5	24.8 ± 1.5
Number of particles compressed	30	30	-

### 2.5. Shell Thickness

Figure 5A displays the cross-section of a LM-microcapsule. There appeared to be a relatively homogeneous shell at point *α* and *β*, and the thickness was 1.25 ± 0.02 µm and 1.33 ± 0.04 µm, respectively. In contrast, point C showed an unclear region, which might result from random slicing at planes other than the equator during sample preparation (artifact) [43,44]. Thirty cross-sections of LM-microcapsules were selected and imaged, which had proven statistically representative [5]. Accordingly, the apparent mean shell thickness (*h*) of LM-microcapsules was determined to be 0.87 ± 0.09 µm which was not statistically different from that of HS-microcapsules (0.78 ± 0.06 µm), as reported in our previous work [24]. The different key mechanical properties between HS- and LM-microcapsules (Table 1) might result from their different structural configuration, and compactness of the shell materials. Furthermore, a strong relationship (*R*^2^ = 0.94) between the shell thickness and the diameter of microcapsules arose (Figure 5B). Interestingly, it was found that *h* increased linearly with the diameter, suggesting that the larger the oil droplet, the thicker the protective shell. 

### 2.6. Adhesive Properties

Fundamental characterisation of the adhesive properties of LM microcapsules onto fabric substrates is vital to understand their potential applications in fabric care via maximising their retention onto target fabrics. A smooth and flat PET film as a model substrate may help to avoid any typical steric effect and physical interference due to crimp and coarse woven fabrics [45]. The retention performance (RP) of LM microcapsules onto PET films as a function of the hydrodynamic shear stress (*τ*) and the Reynolds number (Re) is shown in Figure 6. The corresponding laminar regime within the chamber (0 < *Re* ≤ 30 or 0 < *τ* ≤ 0.9 Pa) is relevant to real washing hydrodynamics, as reported elsewhere [46]. At a low acidic pH 3.2, it was found that the RP of LM-microcapsules was ~70% and ~60% at <0.1 Pa (*Re* < 5) and 0.9 Pa (Re ≃ 30), respectively. In addition, at *τ* > 0.1 Pa the RP of LM-microcapsules was not significantly impaired, and only a marginal reduction at increasingly high Reynolds numbers (5 < *Re* < 30) was observed. This finding suggests that an equilibrium between the two contact surfaces (i.e., microcapsules and PET films) may have been reached. As for pH 3.2, LM microcapsules also exhibited a similar behaviour to that at pH 4.1 where a high RP of ~70% at 50 mPa was recorded. However, unlike pH 3.2, a mild (~5%) reduction in their RP (~55%) at pH 4.1 was observed. Interestingly, at weakly acidic (pH 5.1–6.2) and near neutral (pH 7.2) environments under *τ* ≥ 0.45 Pa (*Re* ≥ 15), the corresponding curve showed a dramatic drop of the RP of microcapsules to 20–30% and <15%, respectively. As can be seen, these values were found to be much lower than those achieved at pH 3.2 and 4.1, where around 60% and 55% of microcapsules had been retained successfully under moderate (0.45 Pa) and high hydrodynamic shear stress (0.9 Pa). At pH 5.1, the RP of LM microcapsules exposed to 0.9 Pa (*Re* ≃ 30) was determined to be ~25%, whereas only <10% of microcapsules had been retained at pH 6.2 and pH 7.2. Overall, the highest RP of LM microcapsules onto PET films was identified at pH 3.2. This finding seems to suggest that a direct effect of the environmental pH on the RP of the microcapsules onto the PET films is significant, regardless of the shear stress applied. Figure 7 displays an example of real-time removal of microcapsules adhering to a fully flat PET film at pH 3.2 under increasingly high shear stress (0–0.9 Pa).

Since LM microcapsules were fabricated via pH-driven CC, their shell may be plausibly pH sensitive. Consequently, the environmental pH may affect the surface charge of microcapsules and the mutual hydrophobic interaction between the shells and PET substrates, hence their adhesive properties. Figure 8 shows the net surface charge (NSC) response of microcapsules with a fCh-GA shell at different pH. As can be seen, a negative trend between the microcapsule NSC and the environmental pH was attained. It was found that the NSC decreased significantly from 15.7 ± 0.4 mV at pH 2 to 3.6 ± 0.1 mV at pH 3 likely due to the increase in OH^−^ in the system. As the pH is further increased to pH 3.4, the isoelectric point (pH_iep_) of the microcapsules was approached with a slightly negative NSC of −0.6 ± 0.1 mV. This result was in line with our previous work elucidating that the optimal pH at which complex coacervation between fCh and GA was driven (pH_CC_ 3.4) effectively led to coacervate shells with almost nil net surface charge [10]. As also documented by Butstraen and Salaün [47], animal chitosan-gum Arabic coacervate shells can be obtained successfully with the strongest electrostatically binding force at pH 3.4–3.6. However, the slightly negative NSC detected at pH 3.4 may be ascribable to the residual ionic strength and/or local ionic imbalances occurring across the electric double layer (Stern and slipping planes) surrounding the microcapsule, which may result in a relative excess of hydroxide ions [48]. As expected, the switchover of the microcapsule surface charge occurred approximately at pH_CC_. As the pH is progressively increased above pH_iep_, more and more negative NSC values were detected. At moderately acidic pH (pH 4–6), the NSC was in the range from −6.4 ± 0.4 mV to −11.3 ± 1.6 mV, whereas at neutral pH, the corresponding NSC was −14.7 ± 1.4 mV. When mirroring the NSC values of LM-microcapsules (Figure 8) with their corresponding RP (Figure 6) at the same pH, it can be noticed that the adhesive properties of the microcapsules onto PET films were maximised around pH_CC_ ∼ pH_iep_. At increasingly high environmental pH values (4.1 < pH ≤ 7.2) their RP was significantly depleted. As reported in literature, untreated PET and polyester fibres are negatively charged at pH 3–9 with a surface charge approximately ranging from −2 mV to −35 mV [49,50]. Therefore, an increasingly negative surface charge of PET with the pH certainly contributed to promoting the repulsive interaction between the microcapsule shell and the PET film since both were highly negatively charged at pH > 4.1. 

Moreover, the effective attachment of LM microcapsules onto the PET may also be related to multiple adhesion mechanisms, such as hydrophobic interaction, hydrogen bonding and Van der Waals forces. When dealing with chitosan, Liu, et al. [51] have also suggested that both amines and carboxyl groups along the backbone of fCh and GA, respectively, were not only responsible for the coacervated shells but also contributed to generating significant hydrogen bonding [52], which may promote the adhesive properties of the resulting microcapsules, especially at low acidic pH. However, it is not clear how the pH affects possible hydrophobic interaction and Van der Waals forces between the microcapsules and PET film, which remains to be investigated in future. 

## 3. Materials and Methods

### 3.1. Materials

Gum Arabic (GA) and fungally fermented chitosan (fCh, degree of deacetylation ~80%) were obtained from Nexira Food (Rouen Cedex, France, EU) and Kitozyme S.A. (Herstal, Belgium, EU), respectively. Analytical grade chemicals including D-limonene (LM; specific gravity at 25 °C ≃ 0.84 g·mL^−1^, CAS number: 5989-54-8), sorbitan triolate (Span85, CAS number: 26266-58-0), triethanolamine (TEA; CAS number: 121-44-8), 50% (*w*/*w*) aqueous glutaraldehyde (GLT; CAS number: 111-30-8), sodium hydroxide (NaOH; CAS number: 1310-73-2), ethanol (CAS number 64-17-5), n-nonane (CAS number 111-84-2) and 36% (*w*/*v*) fuming hydrochloric acid (HCl; CAS number: 7647-01-0) were purchased from Sigma-Aldrich (Gillingham, Dorset, UK), stored according to the Safety Data Sheet (SDS) guidelines, and utilised without additional purification. All the solutions/suspensions were prepared employing demineralised water (MilliQ water, 18.2 MΩ·cm at 25 °C).

### 3.2. Preparation of Microcapsules

The microcapsules were fabricated via one-step complex coacervation according to Baiocco, Preece and Zhang [10]. Briefly, LM (45 mL) was added to an aqueous admixture (130 mL) containing GA (2% *w*/*w*) and fCh (0.3% *w*/*w*) at pH 1.95 which had been acidified by HCl_aq_ (pH probe in solution; FP20, pH resolution ± 0.01, Mettler Toledo, UK), thus affording two phases. Sorbitan triolate (0.8g; surfactant-to-oil weight ratio 0.02) as the emulsifier was added. Emulsification (1000 rpm; 30 min) was carried out immersing a six-blade Rushton turbine (Ø 34 mm) paired with an overhead stirrer (IKA Eurostar 20, Germany, EU) inside a cylindrical jacketed vessel (liquid Height (H)/Tank diameter (T) = 1; T = H = 0.16 m; Clearance (C)/Impeller Diameter (D) ~3/4) with four baffles (baffle width (b)/Tank diameter (T) ~0.1) and thermostated at 25 ± 0.1 °C (Dyneo™ DD-300F, Julabo Ltd., Stamford, UK) to achieve oil-in-water (*o*/*w*) droplets of ~30µm measured by laser diffraction (Malvern Mastersizer 2000, Malvern Instruments, Malvern, England, UK). Complex coacervation between GA and fCh was induced by increasing the pH to 3.4 gradually via addition (160 mL) of TEA_aq_ (30 µL·s^−1^) with a syringe pump (Ultra 70-3007, Harvard Apparatus Inc., Holliston, MA, USA) into the emulsified solution to increase the pH gradually to 3.4 (pH_CC_) under continuous stirring (400 rpm). GLT (0.3 g/g-biopolymer) was added to trigger the crosslinking with amines along the microcapsule shells (reticulation). The suspension of microcapsules was left crosslinking under stirring (300 rpm; ~15 h). The slurry of microcapsules was separated out by vacuum filtration (N938.50KT.18, Laboport Mini Diaphragm Vacuum Pump, KNF Neuberger™, Freiburg, Germany) using membrane filter paper (pore size 0.2 µm; diameter 47 mm, Whatman GmbH, Dassel, Germany, EU), and stored within airtight vials covered with aluminium foil.

### 3.3. Analytical Techniques

#### 3.3.1. Laser Diffraction Particle Size Analysis

Laser diffraction technique was employed to quantify and compare the mean size and size distribution between the emulsion droplets and the resulting microcapsules (Mastersizer2000, Malvern Instruments, UK). The emulsion or microcapsule suspension (~3 mL) was added into the continuously stirred (2500 rpm) sample dispersion unit coupled with the instrument. The tests were performed at ambient temperature by a He–Ne laser with a measurement capability range of 50 nm–0.9 mm. The overall refractive index (~1.5), mean Sauter diameter *D*_[3,2]_ and Span were calculated according to Baiocco, Preece and Zhang [10].

#### 3.3.2. Bright-Field Optical Microscopy

The morphology of the oil droplets and the resulting microcapsules were assessed by bright-field microscopy (Leica DM RBE microscope, Leica Microsystems GmbH, Dassel, Germany, EU). The microscope was equipped with the appropriate magnification lenses (PL Fluotar, 5×/0.12, 506001& 10×/0.30, 506010, 1.0/1.25/1.6×, Leica, Wetzlar, Germany) and paired with an image capturing software package (Motic Images Advanced version 3.2, Motic China Group Ltd., Xiamen, China) which enabled real-time digitalised image capturing of the photomicrographs (Moticam Pro 205B Digital Camera, Leica Microsystems Imaging Solution Ltd., Milton Keynes, UK).

#### 3.3.3. Scanning Electron Microscopy (SEM)

A scanning electron microscope (JEOL 6060, Peabody, MA, USA) was operated at a working distance of 8 mm and accelerated voltage of 15–30 kV to assay the microcapsules for their morphology, surface topography, and structural properties. An aliquot (100 µL) of suspended microcapsules were placed onto an adhesive carbon tab (Leit 9mm disc, Agar Scientific, Stansted, UK) attached to the underlying specimen stub, and left until fully dried. Subsequently, the resulting dry microcapsules were coated with platinum via sputter deposition (Polaron Sputter Coater SC7640, QuorumTech, Sussex, UK) under vacuum (~10^−3^ Pa) and moderate emission current (~25 mA) to achieve a thin metal layer (~8 nm) over 2 min in order to hinder any undesirable charging effect during imaging. Moreover, the SEM micrographs were preliminarily used to estimate the shell thickness of microcapsules which had been cryogenically frozen with liquid nitrogen (−196 °C)_,_ and subsequently ground within an agate mortar and pestle to provoke their mechanical fracture.

#### 3.3.4. Transmission Electron Microscopy (TEM)

TEM micrographs were acquired to image the cross-sectional areas of thirty microcapsules, as well as to measure their shell thickness (JEM-2100 TEM, JEOL Ltd., Tokyo, Japan operating at 200 kV). TEM samples were prepared according to Long, York, Zhang and Preece [42] by embedding microcapsules into hard-grade white acrylic resin inside a centrifugal rotator (4 rpm, 3 min). The resulting cake was left in an oven (60 °C) to set overnight (~15 h). Subsequently, it was sliced into ultrathin sections (~100 nm) by a microtome (UltracutE, Reichert-Jung, Vienna, Austria) prior to fixation onto the carbon coated copper grids (2 mm^2^). The sample was carefully loaded into the TEM stage which was then evacuated (~10^−5^ Pa) to attain the maximal resolution during imaging.

#### 3.3.5. Gas Chromatography (GC)

Gas chromatography separation method was employed to detect GC-amenable LM from liquid samples and quantify its content via standard calibration. Payload (P_%_) and Encapsulation efficiency (EE_%_) were determined according to Baiocco, Preece and Zhang [10]. Air-dried LM-microcapsules (2.5 mg) were suspended in absolute ethanol as the receptor medium (50 mL) within airtight glass vials which were ultrasonicated (VWR Ultrasonicator, USC200TH, Lutterworth, UK) over 60 min to rupture the microcapsule shells hence releasing their oil load. Subsequently, the LM-abundant phase (supernatant) was separated by centrifugation (Sigma 3-18KS, Buckinghamshire, England, UK) and transferred into an automatic parse screw-top amber glass vial (Agilent Technologies, 9-425, CA, US) specially designed for GC. The GC system was a Shimadzu GC-2010, with AOC-20i Injector and AOC-20s Auto-sampler, (Shimadzu UK Ltd., Wolverton, UK) utilising nonane as the internal standard (IS). The column used was a Phenomenex ZB-5 (95% Polydimethyl siloxane/5% Polydiphenyl siloxane) connected to the gas cylinders of He, N_2_, and compressed air as the carrier gases. Based on the response of LM to the specific detector in the column, the elution time of LM was 7.5 min. The experiments were performed in triplicate.

#### 3.3.6. Micromanipulation

The mechanical properties of microcapsules were investigated using a micromanipulation technique based on the parallel plate compression of individual microcapsules [44]. Two droplets (~75 µL/droplet) of LM-microcapsule suspension were pipetted onto a specially designed highly tempered glass slide with a smooth polished flat surface (2.5 cm^2^; length 2.5 cm; width 1.0 cm; thickness 0.3 cm) using a micro spoon spatula, which were left air-drying for 4 h. The glass slide was secured onto a metal stage and placed beneath a borosilicate glass probe (initial diameter 0.1 cm, Harvard Apparatus Ltd., Kent, UK) with a flat-end tip of 60 µm (total length 0.8 cm) which had been polished (bending angle 0°) against a microfine lapping sheet (1 µm) using a rotating micropipette grinder (Model EG-40, Narishige, MicroInstruments Ltd., Long Hanborough, UK). The tip was longitudinally superglued (Loctite Cyanoacrylate Superglue, Henkel, Germany, EU) onto the output borosilicate glass tube (outer diameter 1 mm, inner diameter 0.58 mm; Harvard Apparatus Ltd., Kent, UK) of a selected force transducer head (Series 403A, Model 1568, maximum force scale 5.0 mN, Sensitivity 0.459 mN⋅V^−1^; Aurora Scientific Inc., Ontario, Canada) paired with an electrically powered control box. The force transducer was mounted onto a three-dimensional fine micromanipulator controlled by a servo motor (DC Servocontroller/Driver, CONEX-C, Newport, California, US). The high resolution (±0.2 µm) micromanipulator was operated to move the glass probe upwards/downwards. A descending speed of 2.0 μm·s^−1^ was selected, which allowed the probe tip to contact and progressively compress each single microcapsule. Microcapsules were digitally observed through a high-speed side-view camera (high magnification (10×–140×); 30 fps; AM4023CT, DinoEye C-Mount Camera, Dino-Lite, Hemel Hempstead, UK). A semi-automatic compression mode was set, which enabled the probe to retreat automatically to its default position once the selected microcapsule had been squashed out. Thirty single randomly selected microcapsules were squashed to generate statistically significant results. The compliance of the force transducer probe was recorded three times prior to compressing the microcapsules, and the corresponding mean value was used to quantify the real displacement of the force transducer probe. The used micromanipulation rig is shown in Appendix A.

#### 3.3.7. Zeta Potentiometry

Zeta potentiometry was employed to quantify the net surface charge (NSC) of water suspended microcapsules (0.25% *w*/*w*) with a fCh-GA coacervate shell. A Nano ZS90 (Malvern Instruments Ltd., Worcestershire, UK) was operated at ambient temperature. The equipment software relied on Smoluchowski’s mathematical theory which converts electrophoretic mobility measurements of colloids into zeta (*ζ*) potential or NSC values [10]. Zeta potentiometric titrations were performed utilising 0.1N HCl and 0.1N NaOH to vary the pH of the microcapsule suspension. Specifically, nine specimens at increasingly high pH were titrated to cover the pH range 2.0–10.0, which were stored in airtight glass scintillation vials until zeta potentiometric inspection. Subsequently, the specimens were loaded into disposable polycarbonate-based folded capillary zeta measurement cell equipped with gold-plated electrodes (DTS1070, Malvern Instruments Ltd., Worcestershire, UK) which had been thoroughly pre-rinsed three times using deionised water (18.2 MΩ·cm^−1^) prior to performing each measurement.

#### 3.3.8. Microfluidic Flow Chamber

##### Assembly of a Flow Chamber System

The flow chamber utilised in this study was based on a previous report by Lane, et al. [53]. Its design consisted of two parallel-fitting metal plates (top and bottom plate), which had been manufactured with precipitation-hardened 6061-aluminium alloy (Poynting Physics Workshop, University of Birmingham, West Midlands, UK). A chamber window was built on each plate at the same position. Each window was fitted with pre-cut tempered glass (7.2 ± 0.1 cm × 2.1 ± 0.1 cm; thickness ~0.2 cm), applying cyanoacrylate glass glue (Loctite SuperBonder, Henkel Loctite, CA, USA) to each edge. Silicon O-rings (BS044/046, Polymax, Bordon, UK) were mounted inside to seal the chamber. As detailed by Lane, Jantzen, Carlon, Jamiolkowski, Grenet, Haseltine, Galinat, l. J., Allen, Truskey and Achneck [53], the flow chamber had length and width of 10.5 cm and 1.7 cm, respectively. The resulting central rectangular area (10.5 cm × 1.7 cm) is etched to a depth of 280 µm (flow height), which provides a unique recessed area in the liquid flow path. A fully flat, transparent, smooth, and amorphous PET film (thickness 0.25 mm; Goodfellow Cambridge Limited, Huntingdon, England, UK) was selected as the model fabric. The PET film was cut out into the required rectangular shape (17 ± 0.4 cm × 3.5 ± 0.2 cm) in order to fit in between the top and bottom plates of the chamber. Such configuration allowed to cover the recessed area securely without affecting neither the flow height nor the accuracy of the resulting shear stress which is flow height dependent (Equation (1)). The plates were secured onto each other using ten stainless steel screws (cordless electric screwdriver, AS6NG, Black&Decker, MD, USA) to assure a liquid-tight assembly. Inlet/Outlet soft silicone tubing (inner Ø ~3 mm) was fitted with one-/three-way stopcocks as well as female/male lock-ring luer connectors (Cole Parmer Instruments, England, UK). A luer lock-tip syringe (Fisher Scientific, Loughborough, UK) was employed for sample inoculation, the tip of which was connected to one side of a split-end silicone tubing. The other tubing end was connected to a syringe (60 mL; Becton Dickinson, Spain, EU) loaded with the test solution at the required pH. The syringe was fixed to a syringe pump (Ultra 70-3007, Harvard Apparatus, MA, USA). The full schematic of a flow chamber system is shown in Figure 9, whereas the actual system is displayed in Appendix A.

##### Flow Chamber Methodology

Fluids are fed in and out of the parallel plate flow chamber through the inlet and outlet sections, respectively. Fluids are flowed through the chamber passageways at a given flowrate, thereby generating a shear stress against the walls of the flow chamber. The wall shear stress (*τ_w_*) represents the force per unit area applied to a solid boundary (wall) against a fluid moving in a longitudinal direction to the wall. For a rectangular shaped flow chamber, *τ_w_* can be evaluated as follows:(1)$$\tau_{w}=\frac{\mu \cdot v_{v}}{\alpha \cdot \;\delta \cdot h^{2}}$$
where *μ* [kg·m^−1^s^−1^] is the dynamic viscosity of the fluid, *v_v_* is the flowrate set on the syringe pump [m^3^·s^−1^], *α* = 1/6 is the dimensionless shape factor, *δ* [m] and *h* [m] are the inner width and height of the flow chamber, respectively [53,54,55]. As reported in literature, the flow is expectedly laminar, which can be confirmed by Reynolds number (laminar flow regime if *Re* < 2.1 × 10^3^) for non-circular ducts:(2)$$Re=\frac{\rho_{w}}{\;\mu }\frac{v_{v}}{\delta \cdot h}D_{h}$$
where *ρ_w_* [kg·m^−3^] is the density of the fluid, and *D_h_* is the hydraulic diameter [m] to be evaluated as follows:(3)$$D_{h}=\frac{2}{\frac{1}{\delta }+\frac{1}{h}}$$
which is derived as a ratio of cross-sectional area to the wetted perimeter of the duct.

##### Testing Solution

Test solutions at different pH (3.2, 4.2, 5.1, 6.1, 7.2) were prepared for each experiment to investigate the effect of pH on the retention performance of perfume microcapsules onto PET films. Each test solution was prepared with deionised water by dropwise addition of aqueous 0.1N HCl_aq_ and 0.1N NaOH_aq_ to the required pH.

##### Testing Specimen

Specimens (i.e., diluted suspension of microcapsules to inoculate into the flow chamber) were obtained from microcapsule slurries. Each slurry (~0.05 g) was weighed and added to the required test solution (20 mL) with the desired pH in order to achieve a 0.25% (*w*/*w*) suspension of perfume microcapsules. The resulting specimens were left to rest to equilibrium (30 min), which is similar to the detergent dissolution at the beginning of a real washing cycle. Each specimen was gently shaken prior to its inoculation into the chamber. The size distribution of microcapsules was also measured to prove their size suitable for the height of the passageway (microcapsule-to-height ≤ 0.3) so that no aggregates formed.

##### Investigation of Microcapsules Deposition and Retention onto PET Films

One of the most relevant parameters is the relative density (*ρ_r_*) of microcapsules in aqueous media (*ρ*_w_). It is essential to know whether the microcapsules naturally float (*ρ_r_* < *ρ_w_*) or settle down (*ρ_r_* > *ρ_w_*) in order to configure the flow chamber correctly. For microcapsules with a specific gravity greater than water (*ρ_r_*_,oil_ > *ρ_w_*), the flow chamber should be positioned standardly, with the top plate upwards and facing the microscope lens. However, the specific gravity of LM-entrapping microcapsules is lower than water (*ρ_r_*_,LM_ < *ρ_w_*), thus the flow chamber should be positioned reversed, with the top plate downwards and facing the microscope light. The morphology of microcapsules, their deposition onto PET model films, as well as the topography of the PET model film were assessed by bright-field microscopy. Specifically, the key steps of the process are summarised herein:i.**Loading**: The test solution at the required pH was infused into the chamber with a syringe pump (5 mL·min^−1^; 3 min). The outlet of the flow chamber was slightly tilted upwards to achieve no air bubbles within the system. The chamber was then secured onto the flat microscope stage, and configured downwards according to *ρ_r,_*_LM_;ii.**Image focusing**: The PET film was adjusted into focus onto the lower surface of the PET film in agreement with *ρ_r,_*_LM_. The focus included the cross-sectional area of the chamber;iii.**Deposition:** A specimen aliquot (4 mL) containing suspended microcapsules was inoculated into the flower chamber. Microcapsules were left to rest to equilibrium (30 min), which was monitored via digitalised optical microscopy;iv.**Cleaning:** The test slurry at the required pH and flowrate (15 μL·min^−1^) was fed into the system (5 min). This step helped to remove free oil droplets, which may be due to microcapsules rupturing when inoculated;v.**Flushing:** Increasing flowrates were selected, leading to a gradual increase of the shear stress. Each flowrate (0.065, 0.125, 0.3, 0.6, 1.2, 3.0, 4.5, 6.0, 9.0, 12.0 mL·min^−1^) was held for 3 min to afford the equilibrium between microcapsules and the PET film. Accordingly, images of microcapsules adhering to the PET film were captured every 3 min from a fixed position via bright-field optical microscopy.

#### 3.3.9. Image Analysis

Images taken during flushing were analysed using an open-source image analysis (IA) software package being ImageJ (1.53c, National Institute of Health, Bethesda, MD, USA). *Ad-hoc* coding was developed to automatically analyse the images (supporting information), which enabled to evaluate the specific area ($\hat{a}$) occupied by microcapsules as compared to their background. Images were first fully desaturated to grey scale, and then converted into black & white to minimise any shading effect. Any hollow hole was automatically filled up for LM-microcapsules to appear as black spots on a white background. The area covered by each microcapsule (*A_k_*) was quantified by detecting the diameter of its corresponding black spot. The total number of spots allowed to determine the cumulative area (*A_c_*) actively covered by all microcapsules on each image, hence$\;\hat{a}$:(4)$$\hat{a}=\frac{A_{c}-A_{b}}{A_{d}-A_{b}}=\frac{1-\frac{\sum_{k=1}^{n}A_{k}}{A_{b}}}{1-\frac{A_{d}}{A_{b}}}$$
where *A_d_* and *A_b_* are the maximised area occupied by microcapsules at nil shear stress and the baseline area of commercially available model films (where defects might occur), respectively.

#### 3.3.10. Statistical Analysis

Three independent replicates of each flow chamber experiment were performed unless indicated otherwise. The difference among the results was expressed as mean value ± standard error (St.E).

## 4. Conclusions

The fabrication of LM-laden microcapsules within a vegetable microplastic-free fCh-GA shell via pH-driven complex coacervation was investigated. Spherical as well as slightly elongated microcapsules were attained with a mean Sauter diameter of diameter of 38 ± 2 µm. The highest EE of the microcapsules was ~45%, which is similar to those in literature for similar microcapsules via CC from animal-derived Ch and other biopolymers [56]. The mechanical properties were quantified by a micromanipulation technique. The micromanipulation results highlighted that the rupture force of the microcapsules increased with the diameter, whereas the nominal rupture stress decreased with the size on average, which is similar to other microcapsules reported in literature. However, the formed microcapsules with a core of LM were found to be mechanically weaker than those of vegetable chitosan microcapsules with a core of HS. The mean rupture tension of LM-laden microcapsules was 37.7 ± 3.0 N/m, which was significantly lower than that of HS-microcapsules (71.6 ± 3.9 N/m). Similarly, the comparison between their average nominal rupture stress suggested that LM-laden microcapsules (2.1 ± 0.2 MPa) were also lower than the HS-laden microcapsules (3.6 ± 0.3 MPa). Besides, the microcapsules exhibited an average displacement at rupture of 3.4 ± 0.4 µm, which is approximately half of that of HS-microcapsules (6.3 ± 3.1 µm). In addition, TEM analysis suggested that the shell thickness of LM-laden microcapsules increases linearly with their diameter with an average value of 0.87 ± 0.09 µm, which appeared to be statistically equal to that of HS-microcapsules (0.78 ± 0.06 µm). In other words, LM-laden microcapsules averagely ruptured at a smaller force and strain than HS-microcapsules suggesting the occurrence of less robust shells. This phenomenon may be due to the LM-fChGA interfacial energy level being relatively higher than that of HS-fChGA, possibly leading to microcapsule shells with a less energetically favourable configuration owing to the intermolecular weakening of the solid fChGA particles forming the shell. Under wet conditions, the adhesive properties of LM-laden microcapsules were investigated by a microfluidic device fitted with PET films. The highest retention performance (~60–70%) of LM microcapsules onto PET films was identified at pH 3.2 which is very similar to pH_CC_ 3.4 and the isoelectric point (pH_iep_) of the microcapsules. It seemed that such pH may promote a strong mutual electrostatic interaction and hydrogen bonding between the PET film and the coacervate shells for the investigated hydrodynamic shear stress (0–0.9 Pa). Moreover, it was found that net surface charge of LM-laden microcapsules became more and more negative as the pH was increased, which contributed towards promoting the repulsive interaction between the microcapsule and the PET films already occurring with a negative charge, resulting in a reduced microcapsule retention [49]. Overall, the results herein presented suggest that microcapsules with a core of limonene may offer an opportunity towards more environmentally and user-friendly applications in many FMCG formulations (e.g., fabric care), as well as tackling the ongoing global issues concerning animal-sourced, and microplastic-based products.

## Figures and Tables

**Figure 1 molecules-27-07215-f001:**
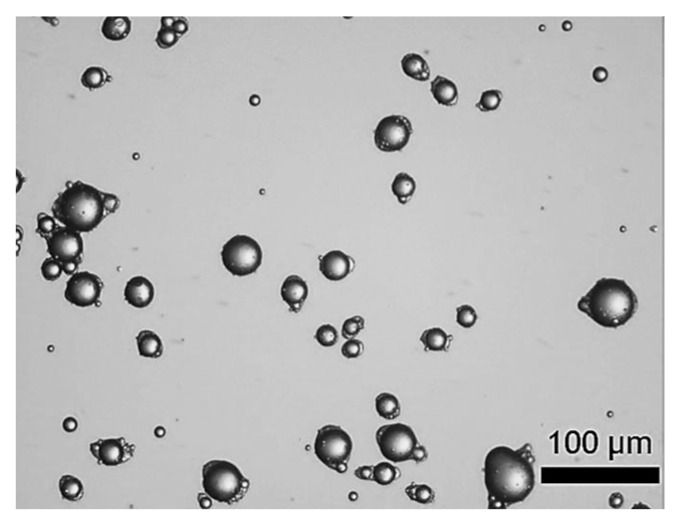
LM-entrapping microcapsules with an elongated eye-like shape by complex coacervation.

**Figure 2 molecules-27-07215-f002:**
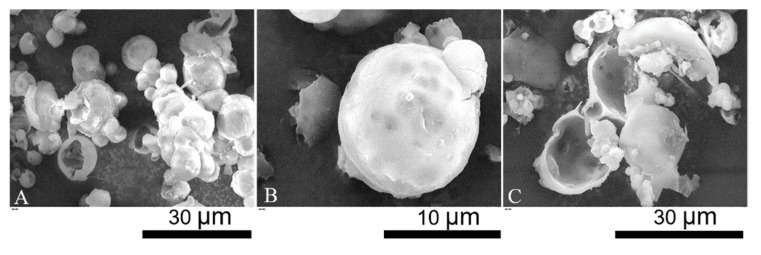
SEM micrographs: overview of single/clusters of microcapsules (**A**); spherical microcapsule with subshell oil (**B**); incomplete microcapsules and polymeric debris (**C**).

**Figure 3 molecules-27-07215-f003:**
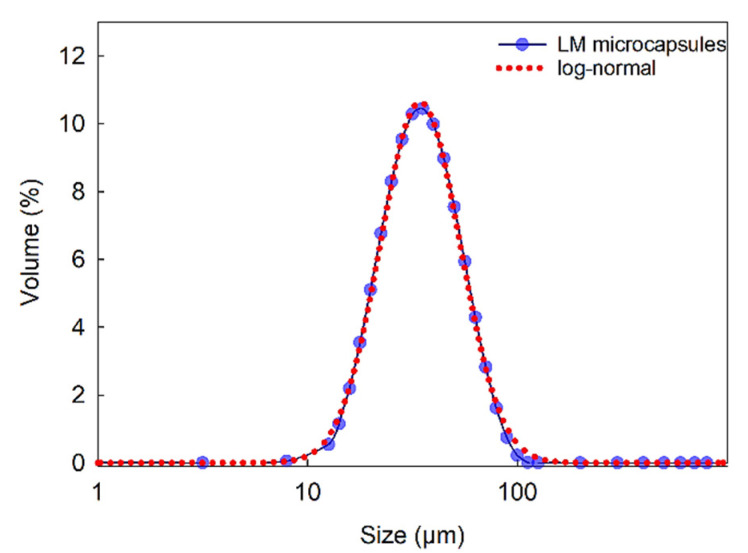
Size distribution of LM-microcapsules fitted to a log-normal distribution function $f\left(x\right)=\frac{a}{x}\exp\left(-\frac{1}{2}{\left(\frac{\ln\left(x/x_{0}\right)}{b}\right)}^{2}\right)$ where *x* is the microcapsule size (diameter) with three parameters *a* = 4.0 × 10^2^; *b* = 0.44; *x_0_* = 41.8 (*R^2^* = 0.998).

**Figure 4 molecules-27-07215-f004:**
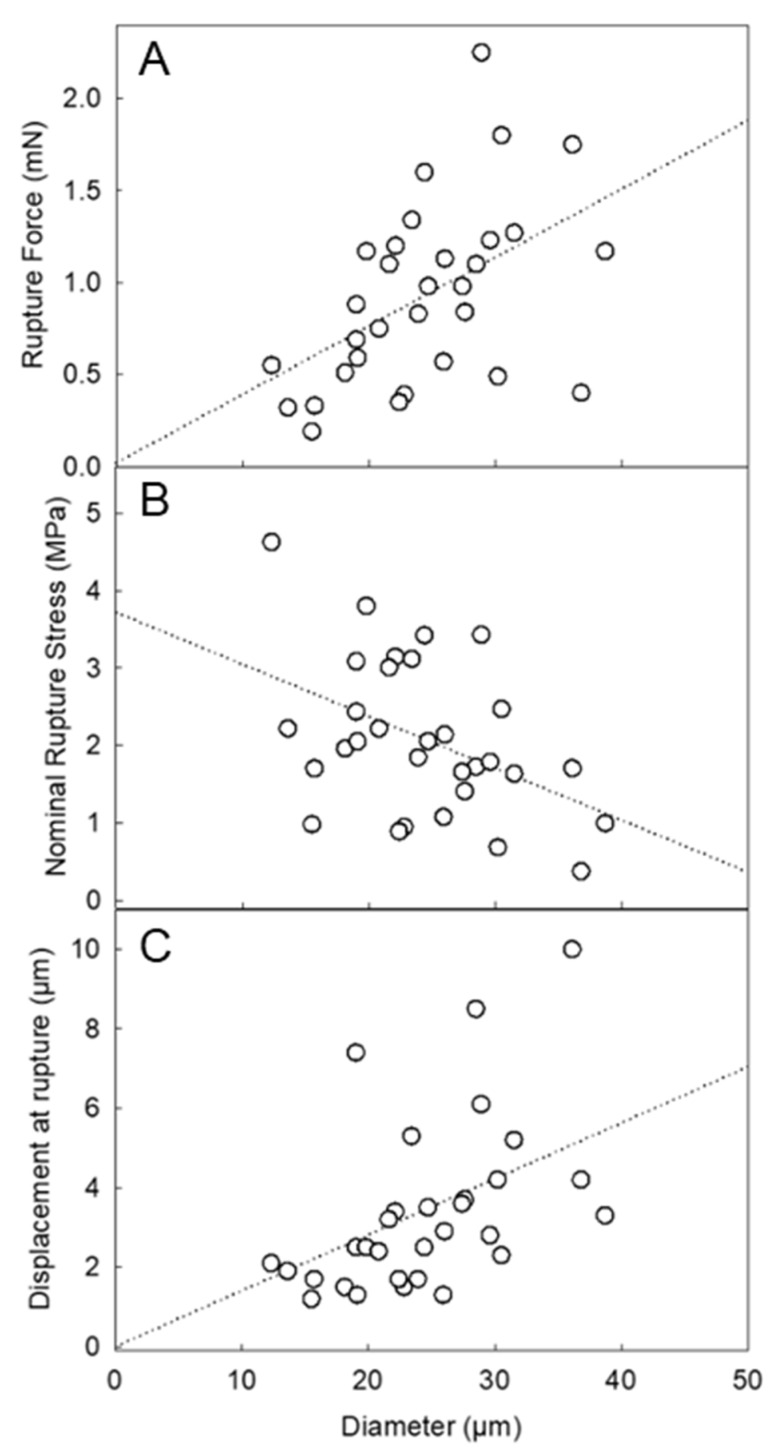
Mechanical property parameters of LM-microcapsules vs. diameter: rupture force (**A**), nominal rupture stress (**B**), and displacement at rupture (**C**). The dotted lines represent the trends only which are simply indicated by a linear fitting of the data in Sigma Plot 14.5 (Alfasoft, Gothenburg, Sweden, EU).

**Figure 5 molecules-27-07215-f005:**
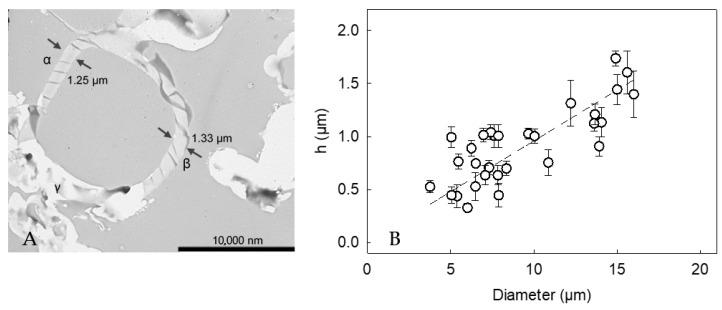
(**A**) Cross-section of a LM-microcapsule (15µm) displaying its shell thickness: (*α*) *h*_A_ = 1.25 ± 0.02 µm and (*β*) *h*_B_ = 1.33 ± 0.04 µm, (*γ*) possible TEM artifacts due to random slicing at planes other than the equator; (**B**) Shell thickness (*h*) *versus* microcapsule diameter. Linear model performance *R*^2^ = 0.94. Error bars may be smaller than the size of the symbols.

**Figure 6 molecules-27-07215-f006:**
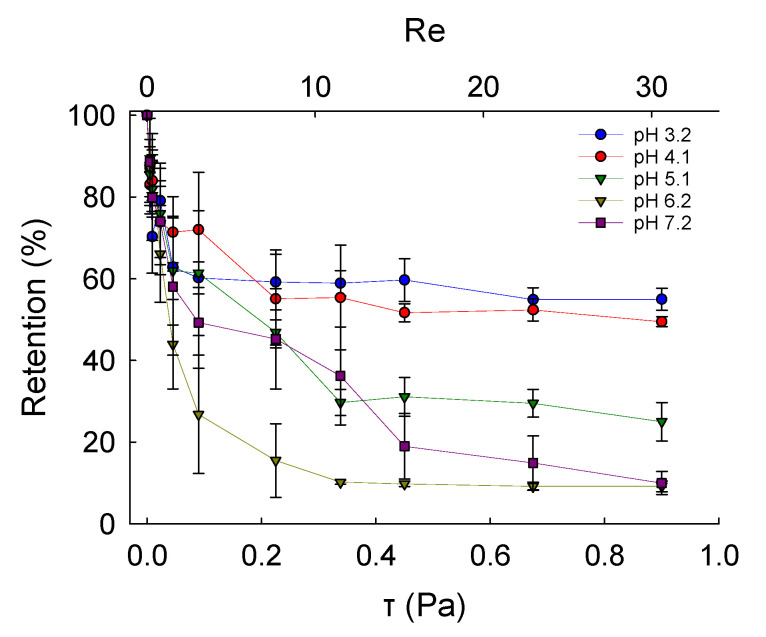
Retention performance of LM microcapsules onto thin PET model films as a function of shear stress (*τ*) and Reynolds number (*Re*; secondary *x*-axis) at different environmental pH.

**Figure 7 molecules-27-07215-f007:**
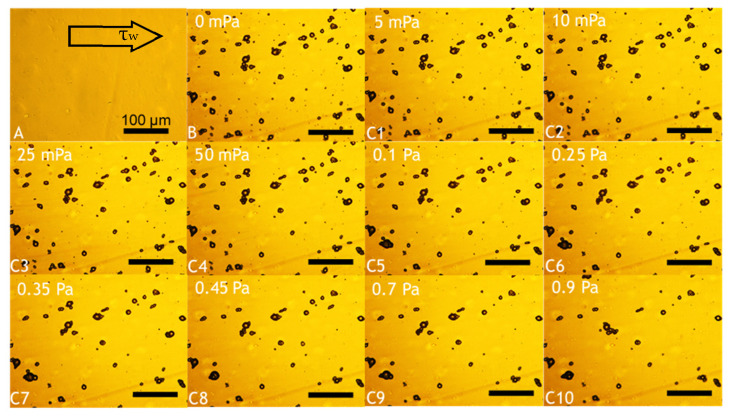
Example of real-time removal of microcapsules adhering to a fully flat PET film at pH 3.2. Background obtained from a blank PET film (baseline) prior to microcapsule inoculation (**A**); deposition of microcapsules following their inoculation and cleaning (**B**); and flushing of microcapsules exposed to increasing shear stress, as follows: 5 mPa (**C1**), 10 mPa (**C2**), 25 mPa (**C3**), 50 mPa (**C4**), 0.1 Pa (**C5**), 0.25 Pa (**C6**), 0.35 Pa (**C7**), 0.45 Pa (**C8**), 0.7 Pa (**C9**), 0.9 Pa (**C10**). The scale bar displayed in (**A**) is applicable to all images.

**Figure 8 molecules-27-07215-f008:**
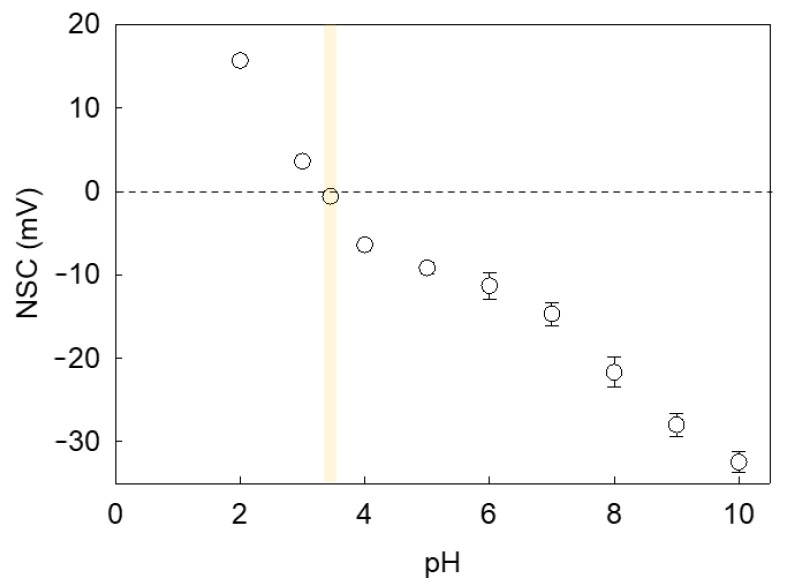
Zeta Potential or Net Surface Charge (NSC) of microcapsules with a fCh-GA coacervate shell exposed to aqueous environments at increasingly high pH values. The vertical yellow strip highlights the isoelectric point of the microcapsules. Error bars may be smaller than the size of the symbols.

**Figure 9 molecules-27-07215-f009:**
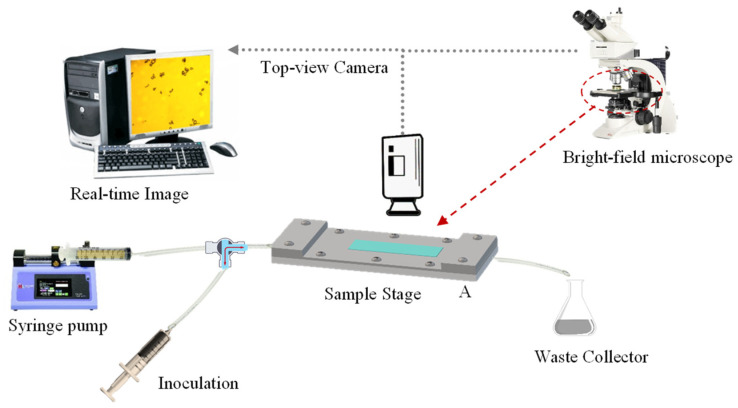
Schematic of a flow chamber system with a fully integrated bright-field optical microscope and computer workstation.

## Data Availability

Not applicable.

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
