# Peer review of "Microplastic-Free Microcapsules to Encapsulate Health-Promoting Limonene Oil"

_molecules, 2022, doi:10.3390/molecules27217215_

Round 1
Reviewer 1 Report
This study consisted of three main objectives a). develop bio-based microcapsules (alternative to plastic) for Limonene Oil, b). mechanical properties of microcapsules were characterized using an advanced micromanipulation technique, and c). adhesive properties were tested using a bespoke microfluidic device. The study was properly planned and executed to obtain the results. Moreover, this is a new material-related lab-scale scientific research study (and has not considered a real application situation, which is not necessary at this stage). Figures/graphs tables were properly prepared.
Some comments:
1. Line 361: "... and Reynolds ----- (Re; secondary x-axis)". "Number" is missing here.
2. Line 87: "potential performance at end-use applications" - Considering the application of used capsules as a material for another product, the reviewer like to propose using either "end-of-life use" or "send-life" rather than end-use (looking at waste management/circular economy perspectives).
3. Table 1: Propose to add a title to the first column (i.e. mechanical property parameter). Remove the reference raw at the bottom and include the citation to the corresponding column title raw.
4. Figure 2: please enlarge the scales.
5. Figure 3: lognormal curve is difficult to observe.
6. Pelase add photographs of the experimental devices as a supplementary document.
Reviewer 2 Report
“Microplastic-free Microcapsules to Encapsulate Health-Promoting Limonene Oil”.
Abstract:
The Abstract is well written, and explain correctly the advances achieved in the researching work and the problematic studied.
Introduction:
Line 69-72:
“Notwithstanding, recent research has evidenced the ubiquitous presence of microplastics in blood as well as animal tissues, and seashells indeed. This type of contamination may also pose a severe risk to humans once microplastic-integrated seafood has entered the human food chain.”
Demineralization of crustacean shells is usually carried out with dilute HCl solutions at room temperature, and Deacetylation of chitin to produce chitosan is usually achieved by hydrolysis of the acetamide groups with concentrated NaOH or KOH (40–50%) at temperatures above 100 °C.
It was demonstrated the presence of microplastics in chitosan obtain from crustacean shells? Please add the reference.
Add a sentence explaining the main chemical differences between chitosan of fungal origin and chitosan derived from seafood, if they exist.
It could be interesting to add a final sentence in the introduction section with the main results of this research work.
Results and Discussion:
Line 107: “Figure 1 displays the digitalised photomicrographs”
How were done these photomicrographs?
Line 108: Please define in the text the acronym CC.
Figure 3: Change the title of the graphic figure, the title cannot be that long.
Figure 4. Mechanical property parameters of LM-microcapsules vs diameter: rupture force (A), nominal rupture stress (B), and displacement at rupture (C). The dotted lines represent the trends only´
For the nominal rupture stress, the trend does not seem to be the one expressed by the dotted line. How were these trends measured?
Figure 5: Change the title of the graphic figure, the title cannot be that long
Conclusions:
Do not use references in the conclusions.
It is necessary to summarize the conclusions with the main results achieved in the research work carried out.
Reviewer 3 Report
The article “Microplastic-free Microcapsules to Encapsulate Health-Promoting Limonene Oil” presents results about the development of safe microcapsules with a core of LM fabricated via complex coacervation using gum Arabic (GA) and fungally sourced chitosan (fCh) as shell precursors. As this study continues authors’ research concerning entrapping relevant substances in microcapsules with natural microplastic-free shells, and the work presents the characterization of their mechanical and adhesive properties to understand their potential performance at end-use applications, in addition, to being well performed with a clear exposition, I recommend its publication in Molecules. Some minor points to address in a revised version are:
i) Could the authors indicate something about the LM leakage rate? This information has important value to evaluate the shelf-life of the product.
ii) I think that in the section Author Contributions there is no need to include the phrase: CRediT Authorship Contribution State
iii) In the Acknowledgments section there are some instructions that need to be deleted: In this section, you can acknowledge any support given which is not covered by the author contribution or funding sections. This may include administrative and technical support, or donations in kind (e.g., materials used for experiments).
